# MR-Guided Radiation Therapy for Prostate and Pancreas Cancer Treatment: A Dosimetric Study Across Two Major MR-Linac Platforms

**DOI:** 10.3390/cancers17162708

**Published:** 2025-08-20

**Authors:** Huiming Dong, Jonathan Pham, Michael V. Lauria, Caiden Atienza, Brett Sloman, Paul Barry, Jennifer Davis, Michael Saracen, Amar Kishan, Ann Raldow, X. Sharon Qi, Daniel Hyer, James Lamb

**Affiliations:** 1Department of Radiation Oncology, University of California, Los Angeles, CA 90095, USA; jonathanpham@mednet.ucla.edu (J.P.); mlauria@mednet.ucla.edu (M.V.L.); aukishan@mednet.ucla.edu (A.K.); araldow@mednet.ucla.edu (A.R.); xqi@mednet.ucla.edu (X.S.Q.); jlamb@mednet.ucla.edu (J.L.); 2Department of Radiation Oncology, University of Iowa, Iowa City, IA 52242, USA; caiden-atienza@uiowa.edu (C.A.); daniel-hyer@uiowa.edu (D.H.); 3Elekta USA, Atlanta, GA 30346, USA; brett.sloman@elekta.com (B.S.); paul.barry@elekta.com (P.B.); jennifer.davis@elekta.com (J.D.); mike.saracen@elekta.com (M.S.)

**Keywords:** MR-guided radiation therapy, MR-linac, dosimetric comparison, prostate cancer, pancreas cancer, SBRT

## Abstract

Magnetic resonance–guided radiation therapy (MRgRT) integrates MR imaging with precise radiation delivery, enabling improved soft-tissue visualization and online adaptive planning. Elekta Unity and ViewRay MRIdian are two leading MR-linac platforms with distinct designs that may affect treatment dosimetric characteristics. This study compared prostate and pancreas cancer treatments by retrospectively re-creating 20 MRIdian clinical cases on the Unity system under MIRAGE and SMART trial protocols. Plans were matched for imaging, contours, beam geometry, and dose prescription to ensure consistency. Target coverage, organ-at-risk (OAR) doses, and treatment times were assessed. Both systems achieved comparable plan quality and efficiency, with Unity showing modest OAR dose reductions. These findings highlight the dosimetric implications of MR-linac design differences and support informed clinical and technological decision-making.

## 1. Introduction

MR-guided radiation therapy (MRgRT) is an important and rapidly evolving modality for delivering adaptive radiation therapy (ART). Compared to conventional computed tomography (CT)-based external beam radiation therapy (EBRT), MR imaging (MRI) uniquely provides excellent soft tissue contrast that allows accurate target and organs-at-risk (OARs) delineation in many sites such as the prostate and pancreas [1,2,3,4]. In addition to anatomical imaging, biomarkers derived from advanced MRI techniques [5,6,7,8,9,10,11,12,13] have a clinical potential to probe various aspects of the tumors and the surrounding normal tissues, providing rich functional information that is rarely available via conventional CT [14]. Moreover, by taking advantage of non-ionizing real-time cine imaging, MRgRT is able to track the target in real-time and gate the treatment beam, ultimately minimizing the effect of intrafraction motion and allowing for a reduction in the PTV margin, potentially decreasing normal tissue toxicity [15,16]. When synergizing all the strengths of MRI with a workflow that integrates online plan adaptation, MRgRT has become a powerful tool for online ART and has attracted much interest from the radiation oncology community.

The Elekta Unity system (Elekta AB, Stockholm, Sweden) and ViewRay MRIdian system (ViewRay Systems Inc., Oakwood Village, OH, USA) are two widely recognized MR-linac platforms that will be compared in this work. With many institutions considering starting an MRgRT program, a common question arises: Do both systems provide comparable treatment quality? Despite sharing the same basic concept of MRgRT, technical differences between the design of the two systems are non-trivial [17]. For example, the Unity system uses a 1.5T magnet, whereas the MRIdian system has a 0.35T magnet. This may give rise to different levels of electron return effect and thus result in different dosimetric characteristics during treatments [18]. In terms of multi-leaf collimators (MLCs), the two systems selected different designs, which can also potentially lead to different dosimetric behaviors. A double-stack, double-focused MLC with 8.3 mm leaf width was employed in the MRIdian system to yield an effective leaf width of 4.15 mm at isocenter. On the other hand, the Unity system utilizes a single-focused MLC with a leaf width of 7 mm and a virtual leaf width of 1 mm achieved through the orthogonal dynamic jaws. Table 1 lists the primary differences between the two platforms that can affect the quality of a treatment plan.

Both the prostate and pancreas are among the most commonly treated anatomical sites on MR-linac platforms, largely due to the superior soft-tissue contrast provided by MRI, as well as the integration of real-time gating and online adaptive capabilities. These features are particularly advantageous for managing organ motion and minimizing dose to adjacent critical structures during prostate and pancreas treatment, allowing for toxicity reduction and dose escalation. The MRIdian system has been employed in two major clinical trials studying the impacts of hypofractionation with MRgRT. The MRI-guided stereotactic body radiation therapy (SBRT) for prostate cancer (MIRAGE) trial studied the differences in toxicities in prostate patients treated with MRgRT using 2 mm PTV margin versus a conventional CT-guided EBRT using 4 mm PTV margin [15], demonstrating reduced acute and late genitourinary (GU) and gastrointestinal (GI) toxicity [19]. The stereotactic magnetic resonance-guided adaptive radiation therapy (SMART) trial investigated acute toxicity in patients with pancreatic cancers treated with dose-escalated MR-guided online ART and showed no acute grade ≥ 3 GI toxicity [20]. It remains an important question whether the Unity system delivers comparable dosimetric performance in these two practice-defining trials.

With these pivotal trials in mind, the design of this study is to evaluate the dosimetric differences between these two MR-linac systems in prostate and pancreas cancers. Dosimetric characteristics of the two MR-linacs were compared via re-creating the treatment plans retrospectively on the Unity system for 20 clinical prostate and pancreas patients previously treated on the ViewRay MRIdian platform at our institution under the MIRAGE and SMART trial constraints, respectively. Finally, the deliverability of the Unity was investigated through IMRT verification QA on a Unity MR-linac system. We hypothesize that Unity can produce treatment plans with comparable quality to those of MRIdian.

**Table 1 cancers-17-02708-t001:** Main Differences between MRIdian and Unity Systems.

	ViewRay MRIdian	Elekta Unity
Imaging Magnet	0.35T	1.5T
Beam Quality	6MV FFF	7MV FFF
Nominal Dose Rate at Dmax	600 MU/Min	535 MU/Min
Source-Axis Distance (SAD)	90 cm	143.5 cm
Magnet configuration	Split bore magnet, radiation therapy beam passes between magnets	Single bore magnet placed co-axially inside the gantry using single cryostat with low attenuating region between superconducting coils to maximize magnetic field homogeneity
Field Shaping	Double focus, double stack primary MLC with 4 mm effective leaf width	Single focus MLC, 7 mm leaf with orthogonal dynamic jaws to produce virtual leaf width of 1 mm
Planning and Optimization	Dose-Volume-Based Optimization	Multi-Criteria Optimization Including Biological Modeling of Tissues [21]
Dose Calculation Algorithm	Voxel-Based Monte Carlo (VMC) [22]	GPU Monte Carlo Dose Calculation (GPUMCD) [23]
Field size	27.4 cm × 24.1 cm	57.4 cm × 22 cm

## 2. Materials and Method

### 2.1. Clinical MRIdian Treatment

Twenty patients previously treated on MIRAGE or SMART trials using the ViewRay MRIdian system at our institution from 2021 to 2023 were randomly selected (*n* = 20). The exclusion criterion was a different prescription dose than the one recommended by the respective trials. Ten were treated for prostatic adenocarcinoma under the MIRAGE trial dose constraints (*n* = 10), and the other ten were treated for pancreatic adenocarcinoma under the SMART trial dose constraints (*n* = 10). It is important to note that the MIRAGE trial is non-adaptive, whereas the SMART trial is. Thus, only the initial reference treatment plans were investigated for the SMART trial in this work.

All MRIdian treatments were planned by board-certified medical dosimetrists and/or board-certified medical physicists in the ViewRay treatment planning system (TPS) on the on-board MR simulation images acquired using balanced steady-state gradient-recalled echo (True FISP) sequence. The planning process followed both the site-specific trial dose constraints as well as our institutional guidelines. Electron density maps were derived from the CT simulation scans and were deformably registered to the MR simulation scans for dose calculation.

### 2.2. Unity Treatment Plans

All Unity treatments were re-created by medical physics residents under the supervision of board-certified medical physicists using the Monaco treatment planning system (Version 5.59.11, Elekta Inc., Stockholm, Sweden). For each patient, the same simulation MR images, CT images, structure set, number of beams, and beam orientations were maintained for both systems, except when a beam angle was blocked by the cryostat pipe of the Unity system [24]. In those cases, minor beam angle adjustments were made to avoid the cryostat pipe. For treatment optimization and dose calculation, the maximum number of segments and dose calculation grid sizes were also kept the same for both MRIdian and Unity plans. Specifically, a dose calculation grid of 2 mm and 3 mm was employed for the prostate and pancreas treatments, respectively. The 3 mm dose calculation grid was used in the clinical pancreas plans to accelerate the online plan adaptation process for shortening the wait time when the patient was on the treatment couch. Electron density maps were obtained via bulk assignment in the Unity plans. Relative electron density of 1 and 1.22 was used for the soft tissue and bone structures, respectively. Both systems calculated the dose to medium [25]. Table 2 summarizes the relevant plan parameters. When optimizing the Unity treatment plans, the plan quality reports (PQRs) of the corresponding MRIdian plans were available to the planners. To evaluate the plan quality, the Unity plans were normalized to the same target coverage as the MRIdian plans.

### 2.3. Unity Deliverability, Monitor Unit, and Delivery Time

The deliverability of all Unity treatments was investigated by calculating the treatment plans for a model of the ArcCHECK-MR (Sun Nuclear, Melbourne, FL, USA) and delivering the IMRT QA plans on a Unity MR-linac system. Subsequently, Gamma analysis was performed to compare the measured dose to the planned dose in absolute dose mode. The criteria of the analysis were dose difference = 3%, distance to agreement (DTA) = 2 mm, and a threshold of 10% of the global maximum dose as recommended by AAPM TG-218 [26]. MRIdian plans were previously delivered clinically and subject to clinical quality control procedures, the deliverability of these plans did not need to be assessed for this study.

The Unity treatment monitor units (MUs) and delivery times were compared to their MRIdian counterparts. For each Unity plan, the delivery time was measured while performing IMRT QA on the system. The estimated delivery times from the MRIdian TPS were then compared to the measured delivery times of the Unity plans. This comparison was considered rational after confirming that the MRIdian’s estimated delivery times were on average 1.56 min (Range: 1.26–1.87 min) shorter than the actual delivery times measured after delivering one prostate and two pancreas plans of this study on the MRIdian system.

### 2.4. Statistical Analysis

Data were presented as mean ± SDs. Data normality was first examined via Shapiro–Wilk test. Paired *t*-test or the Wilcoxon rank sum test was then performed to compare the difference in the OAR dose between the MRIdian and Unity plans. A *p*-value < 0.05 was considered statistically significant.

## 3. Results

### 3.1. Treatment Plan Quality Comparison

Table 3 and Table 4 summarize the dosimetric comparison between MRIdian and Unity for prostate and pancreas plans, respectively.

For the prostate patients, no significant difference was observed in the %target volume receiving 42 Gy or higher (trial goal: ≤30%, difference = 2.02%, *p* = 0.427) when comparing Unity to the MRIdian counterparts. However, Unity had lower rectum V36Gy (trial goal: ≤10%, difference = 0.52%, *p* = 0.0095), V38Gy (trial goal: ≤5%, difference = 0.31%, *p* = 0.0043) and V40Gy (trial goal: ≤2%, difference = 0.26%, *p* = 0.0469). Additionally, significantly lower left (trial goal: ≤5 cc, difference = 1.18 cc, *p* = 0.0137) and right femur V20Gy (trial goal: ≤5 cc, difference = 0.96 cc, *p* = 0.0020) were also observed in the Unity plans. No significant dose difference was observed in the other prostate OARs. Figure 1 demonstrates the dosimetric comparisons in OARs that are statistically different.

For pancreas patients, no significant dose difference was observed for the OARs except for the liver when comparing Unity to MRIdian. The Unity treatments had significantly lower mean liver dose (trial goal: ≤20 Gy, difference: 1.40 Gy, *p* = 0.0371). Figure 2 demonstrates the dosimetric comparisons between the two systems in critical mucosal tissues near the treatment target.

Figure 3a compares the target dose distributions of a representative prostate and pancreas case. Isodose lines of 50% and 100% of the prescribed dose were displayed. Figure 3b,c evaluate the cumulative dose-volume histograms (DVHs) of the two aforementioned prostate and pancreas cases.

### 3.2. Unity Deliverability, MU, and Delivery Time

All treatment plans have a Gamma passing rate >90% under 3%/2 mm criteria, confirming clinical deliverability of the Unity plans. The mean Gamma passing rate was 96.93 ± 1.32% (Range: 94.50–98.90%) and 95.51 ± 2.25% (Range: 91.20–98.60%) for the prostate and pancreas plans, respectively.

Figure 4a,b demonstrates the number of monitor units between the MRIdian and Unity plans. The mean number of MUs of the prostate treatments was 3398 ± 791 and 2478 ± 372 for MRIdian and Unity, respectively. The mean number of MUs of the pancreas treatments was 5758 ± 1375 and 4142 ± 1423 for MRIdian and Unity, respectively. Unity used a smaller number of MUs compared to MRIdian for both prostate (difference = 920 MUs, *p* = 0.0036) and pancreas (difference = 1616 MUs, *p* = 0.0076). Figure 4c,d compares the treatment delivery time between the MRIdian and Unity systems. The mean delivery time for prostate treatments was 12.78 ± 1.68 (TPS-estimated) and 13.53 ± 1.88 (measured) minutes for MRIdian and Unity, respectively. The mean delivery time for pancreas treatments was 14.58 ± 2.78 (TPS-estimated) and 17.40 ± 3.77 (measured) minutes for MRIdian and Unity, respectively.

## 4. Discussion

In this study, a dosimetric investigation of two major MR-linac systems was performed by investigating 20 clinical plans previously treated using the ViewRay MRIdian at our institution. The corresponding Unity treatment plans were generated in a Monaco treatment planning system using matching planning images, structure sets, as well as beam geometry and dose calculation parameters, as the clinical MRIdian plans. Similar plan quality was observed between the two systems. The deliverability of the Unity plans was demonstrated through Gamma analysis after successfully delivering all the plans on a Unity MR-linac system.

### 4.1. Prostate

The MIRAGE trial requires the target volume receiving 42 Gy or higher to be lower than 30% of the total target volume (i.e., PTV V42Gy < 30%) [15]. This dose constraint is crucial because it reflects the heterogeneity of the PTV dose. Moreover, for critical organs at risk near or within the prostate, such as the bladder, rectum, and urethra, all have a point dose constraint of 42 Gy, highlighting the importance of reducing the V42Gy in the target. When comparing the treatment plans between the two systems, no significant difference was observed in PTV V42Gy, demonstrating plan equivalency in this regard.

Unity showed statistically significant reductions in rectum and femur dose compared to the MRIdian equivalents. It is important to note that despite the statistical significance, plans from both systems met the OAR dose constraints defined by the MIRAGE trial [15]. Furthermore, the dose differences between the two systems were minimal with respect to the trial-defined constraint values, suggesting that the observed rectum and femur dose variations between the two systems may not directly translate into clinically meaningful differences that can affect treatment outcomes.

None of the prostate patients investigated in this study had nodal disease. Thus, the dose to the sigmoid was minimal or absent in all treatments, regardless of the planning system, as the sigmoid is typically located superior to the prostate. For this reason, no statistical analysis could be meaningfully performed. However, the dose to the sigmoid would be non-trivial and should be carefully evaluated for any plans that include lymph node targets [27,28].

### 4.2. Pancreas

No significant difference in dose to the OARs was observed between the MRIdian and Unity pancreas plans, except for the mean liver dose, which was significantly lower in the Unity treatment plans compared to the MRIdian counterparts. There was greater variation in OAR dose across the pancreas patients than that of the prostate patients in both the MRIdian and Unity treatments. This is likely due to the considerable difference in the relative positions of the OARs with respect to the PTV targets across these individuals, especially in the critical mucosal tissues such as the stomach, duodenum, and bowels. These broad anatomical variations were successfully captured in our study, and the overall difference in pancreas plan quality was negligible between the two planning systems.

A typical dose constraint for the critical mucosal structures in the SMART trial is V33Gy < 0.5–1 cc [20,29]. Our institutional guidelines recommend a V33Gy < 0.5 cc constraint for the initial plan, with allowance for V33Gy < 1 cc during the online adaptation. Meeting the more stringent dose constraint (i.e., V33Gy < 0.5 cc) during the initial planning phase facilitates the achieving of V33Gy < 1 cc in the subsequent online adaptations, thus serving as an indicator of the readiness for plan adaptation. In this study, both planning systems successfully met the 0.5 cc constraint on average with no significant difference, suggesting the equivalency for potential online plan adaptation.

### 4.3. Beam Shaping and Plan Quality

The Unity system elected a single stacked and single focused MLC design, whereas MRIdian utilized a double-stack, double-focused MLC with a smaller effective leaf width and shorter SAD. These differences in beam shaping between the two systems could potentially impact treatment plan quality in terms of dose conformity, reducing beam penumbra, and allowing steeper dose falloff. Interestingly, comparable plans were observed in our study between the two systems. This may be partially attributed to the dynamic jaw tracking of the Unity system, along with the Monaco TPS, which provides a relatively more intuitive planning experience and advanced planning capabilities such as multi-criteria and biology-based optimization, compensating for the differences in MLC and machine SAD.

### 4.4. Treatment MU and Delivery Time

Unity utilized a smaller number of MUs compared to MRIdian for both prostate and pancreas treatments. This was achieved mainly by increasing both fluence smoothing and minimum MU per segment. However, the lower MU did not directly translate into faster delivery times. This may be due to Unity’s relatively lower dose rate of 535 MU/min compared to 600 MU/min of the MRIdian system at Dmax. The delivery efficiency of Unity can be potentially improved via exploring the feasibility of volumetric modulated arc therapy (VMAT) MRgRT, although this functionality is still under active research and has not yet become clinically available [30].

### 4.5. Limitations

There are several limitations in this study. First, the MRIdian and the Unity plans of the same patient were created by different planners. The clinical MRIdian treatments were generated by board-certified dosimetrists and/or physicists, whereas the Unity counterparts were planned by resident physicists after a short course of Monaco TPS training. To account for the difference in planning experience, Unity plans were supervised by three board-certified medical physicists with expertise in MRgRT planning on MRIdian (J.L. and X.Q.) and Unity (D.H.) platforms. Second, the clinical plans were produced under a typical turnaround time of seven days, while no explicit time constraint was imposed when creating the Unity counterparts. Additionally, the plan quality reports of the MRIdian plans were not blinded to the Unity planners, and thus, what each plan could potentially achieve dosimetrically was known to the Unity planners a priori. This knowledge may have guided the approach to generating Unity plans by setting a benchmark, partially contributing to the similar dosimetric performance observed between the two platforms. On the other hand, similar planning parameters were maintained for both systems during the plan optimization. These parameters were optimized for the MRIdian system and, thus, may not fully leverage the capabilities of the Monaco TPS and Unity platform. Nevertheless, the key question investigated in this work was whether the Unity system would yield comparable plans to those of the MRIdian system in the two aforementioned trials [15,20]. Our study successfully demonstrated similar dosimetric performance between the two systems despite the different system designs. Secondly, we only investigated the initial reference treatment plans for each patient. No comparison was made for any treatment adaptations. Hence, the different dosimetric performance of online adaptation remains a question that should be studied in future work. In addition, it is also important to note that motion management during treatment delivery can play a critical role in the final dose received by the targets and OARs. For example, the allowed number and orientation of imaging planes, achievable spatial and temporal resolution, as well as the possibility of beam trailing for baseline motion correction, are different between the MRIdian and Unity motion management systems [31,32,33]. These different gating techniques and motion mitigation strategies can influence the precision of radiation dose delivery to the target while minimizing exposure to the surrounding OARs. Although the differences do not affect the dosimetric characteristics of the initial reference plan and, therefore, do not impact the findings of this study, they may alter the actual dose received by the patients during the treatment, potentially influencing clinical endpoints. Finally, this is a retrospective study with a limited number of patients, potentially limiting the statistical power. Moreover, only prostate and pancreas treatments have been investigated in this work. Evaluation across additional anatomical sites with daily anatomical variation using a larger sample size and clinical endpoints could provide deeper insights into the unique advantages and challenges of each system, and should be studied in the future work.

## 5. Conclusions

In this study, we randomly selected 20 MRIdian patients from the MIRAGE and SMART trials, and retrospectively created their treatment plans on the Unity system. Equivalent plan quality was observed for both prostate and pancreas treatments between the two systems: most clinically relevant dose-volume metrics showed non-statistically significant differences, while differences that were statistically significant were unlikely to be clinically significant. Finally, performing IMRT QA on a Unity MR-linac revealed high deliverability of the Unity plans. These findings suggest that both MR-linac systems offer comparable capabilities for high-quality radiation therapy planning.

## Figures and Tables

**Figure 1 cancers-17-02708-f001:**
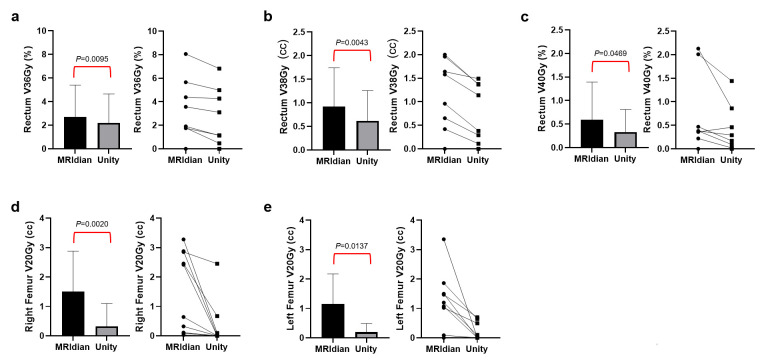
Prostate OAR Dose Comparison between the MRIdian and Unity Systems. Lower rectum V36Gy (**a**), V38Gy (**b**), V40Gy (**c**), as well as right (**d**) and left (**e**) femur V20Gy, were observed with Unity compared to MRIdian. The difference reached statistical significance. Ladder plots were displayed in the figure to demonstrate the paired OAR dose comparison between the two systems of each individual case.

**Figure 2 cancers-17-02708-f002:**
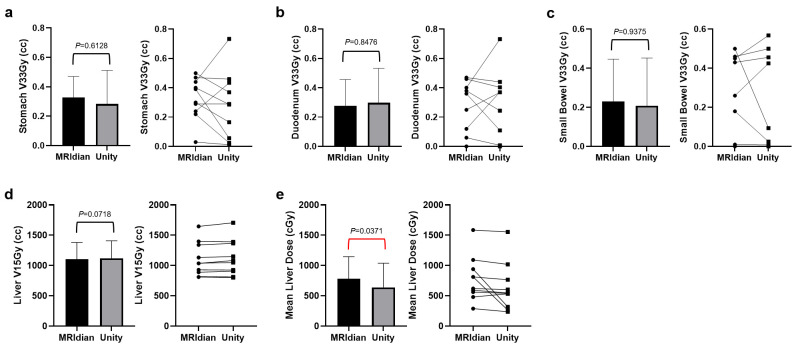
Critical mucosal tissue dose comparison between the MRIdian and Unity systems. The dose comparison in the stomach (**a**), duodenum (**b**), small bowel (**c**), V33Gy, and liver V15Gy (**d**) revealed no significant difference between Unity and MRIdian. Interestingly, in some cases, Unity also demonstrated a relatively large positive or negative dose difference from the MRIdian equivalents in comparison to that of the prostate plans, despite the lack of statistical significance in general, which may reflect the variation in optimization strategies employed by different planners when addressing complex treatments like those in the pancreas patients. Lower mean liver dose was observed with Unity when compared to MRIdian (**e**).

**Figure 3 cancers-17-02708-f003:**
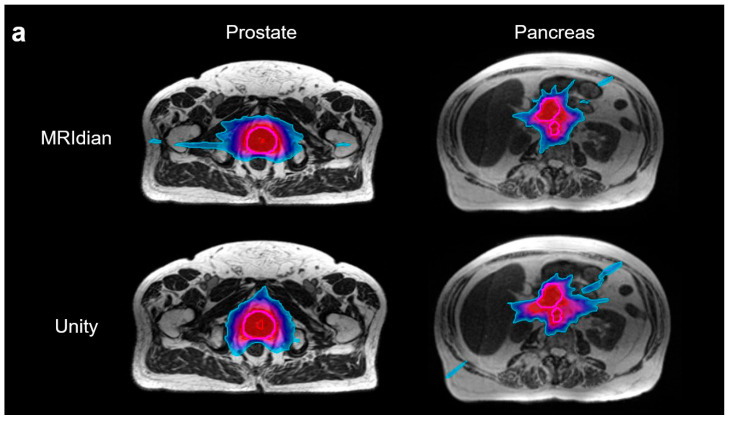
Treatment plan dose comparison. (**a**) Representative dose distributions are demonstrated for 50% (red) and 100% isodose (blue) levels in the prostate and pancreas between the MRIdian and Unity plans. Treating targets was highlighted in magenta. Comparable DVHs were observed across the two systems for targets and OARs in prostate (**b**) and pancreas (**c**). Solid line represents the Unity plans, whereas the dotted line represents the MRIdian plans.

**Figure 4 cancers-17-02708-f004:**
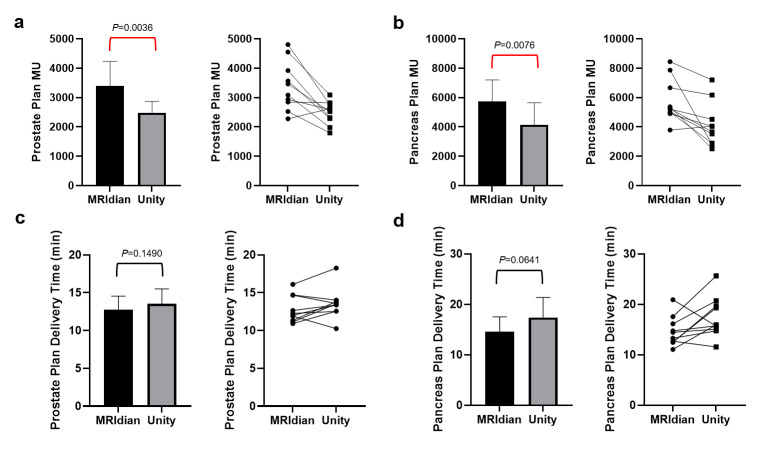
Treatment MU and delivery time comparison between the MRIdian and Unity systems. Lower MUs were used in the Unity treatments vs. the MRIdian treatments for prostate (**a**) and pancreas (**b**). Treatment delivery time estimated by the ViewRay MRIdian TPS was compared to that measured on the Unity system for prostate (**c**) and pancreas (**d**). Note that the estimated delivery times were about 1.56 min shorter than the actual measured delivery times for the MRIdian system.

**Table 2 cancers-17-02708-t002:** Parameter comparison between the MRIdian and unity treatment plannings.

	ViewRay MRIdian	Elekta Unity
Treatment Technique	Step-and-Shoot IMRT	Step-and-Shoot IMRT
No. of Beams	Per Plan (15–21 beams)	Same as the MRIdian Plans
Beam Angles	Per Plan	Same as the MRIdian Plans
Maximum No. of Segments	Per Plan	Same as the MRIdian Plans
Dose Calculation Grid Size	Prostate: 2 mm; Pancreas: 3 mm	Prostate: 2 mm; Pancreas: 3 mm
Dose to Medium	Yes	Yes
Electron Density	Derived from Deformed CT	Bulk Assignment

**Table 3 cancers-17-02708-t003:** Plan quality comparison between the MRIdian and Unity prostate plans.

Prostate SBRT (MIRAGE Trial, 8 Gy × 5 to the PTV and 8.4 Gy × 5 to the GTV, *n* = 10)					
	Trial Goal	MRIdian	Unity	∆(MRIdian−Unity)	*p*-Value
PTV V40Gy (%)	≥95	91.93 ± 3.28	91.93 ± 3.28	0	NA
PTV V42Gy (%)	≤30	21.69 ± 6.77	19.67 ± 6.05	2.02	0.4270
GTV V42Gy (%)	≥95	93.08 ± 11.96	94.87 ± 10.15	−1.80	0.2240
Rectum V20Gy (%)	≤50	15.24 ± 9.26	14.47 ± 9.59	0.76	0.4459
Rectum V24Gy (%)	≤50	10.19 ± 7.79	10.55 ± 7.76	−0.35	0.5732
Rectum V32Gy (%)	≤20	4.86 ± 4.41	4.53 ± 4.25	0.32	0.1978
Rectum V36Gy (%)	≤10	2.71 ± 2.69	2.20 ± 2.45	0.52	0.0095 *
Rectum V38Gy (cc)	≤2	0.92 ± 0.82	0.62 ± 0.64	0.31	0.0043 *
Rectum V40Gy (%)	≤5	0.60 ± 0.80	0.33 ± 0.48	0.26	0.0469 *
Rectum V42Gy (cc)	≤0.035	0.005 ± 0.01	0.003 ± 0.006	0.002	1.0000
Bladder V20Gy (%)	≤40	13.02 ± 10.13	14.06 ± 11.83	−1.05	0.7695
Bladder V39Gy (cc)	≤4	1.00 ± 1.26	1.27 ± 1.77	−0.27	0.8203
Bladder V40Gy (%)	≤5	0.13 ± 0.16	0.08 ± 0.13	0.05	0.1924
Bladder V42Gy (cc)	≤0.035	0.005 ± 0.01	0.0001 ± 0.0003	0.005	0.8203
Anal Canal V20Gy (cc)	≤5	0.30 ± 0.52	0.23 ± 0.24	0.07	0.7695
Anal Canal V30Gy (cc)	≤0.035	0.005 ± 0.008	0.004 ± 0.006	0.001	0.6513
Urethra V42Gy (cc)	≤0.035	0.006 ± 0.01	0.005 ± 0.008	0.0004	0.9179
Femur Right V20Gy (cc)	≤5	1.51 ± 1.37	0.32 ± 0.78	1.18	0.0020 *
Femur Left V20Gy (cc)	≤5	1.16 ± 1.02	0.19 ± 0.29	0.96	0.0137 *
Penile Bulb V24.8Gy (%)	≤5	0.15 ± 0.49	0.06 ± 0.19	−1.02	0.3434
Sigmoid V25Gy (cc)	≤30	0.03 ± 0.09	0.14 ± 0.41	−0.10	NA
Sigmoid V30Gy (cc)	≤1	0	0.05 ± 0.15	−0.05	NA
Sigmoid V38Gy (cc)	≤0.035	0	0	0	NA

* Statistical significance. NA: no meaningful test could be performed due to insufficient number of non-zero difference pairs.

**Table 4 cancers-17-02708-t004:** Plan quality comparison between the MRIdian and Unity pancreas plans.

Pancreas SBRT (SMART Trial, 10 Gy × 5 to the PTV, *n* = 10)					
	Trial Goal	MRIdian	Unity	∆(MRIdian−Unity)	*p*-Value
PTV V50Gy (%)	≥95	80.28 ± 10.28	80.28 ± 10.28	0	NA
Stomach V33Gy (cc)	≤0.50	0.33 ± 0.14	0.29 ± 0.23	0.05	0.6128
Duodenum V33Gy (cc)	≤0.50	0.28 ± 0.18	0.30 ± 0.23	−0.02	0.8476
Small Bowel V33Gy (cc)	≤0.50	0.23 ± 0.22	0.21 ± 0.24	0.02	0.9375
Large Bowel V33Gy (cc)	≤0.50	0.008 ± 0.02	0	0.008	NA
Liver <15 Gy (cc)	≥700	1103.79 ± 276.93	1118.90 ± 291.08	−15.11	0.0718
Liver Mean Dose (Gy)	≤20	7.79 ± 3.67	6.38 ± 3.99	1.40	0.0371 *
Spinal Cord V25Gy (cc)	≤0.50	0	0.0006 ± 0.0019	−0.0006	NA
Kidney Right V14Gy (%)	≤33%	9.243 ± 12.35	11.07 ± 12.88	−1.823	0.73289
Kidney Left V14Gy (%)	≤33%	7 ± 7.59	6.83 ± 8.17	0.1167	0.41937
Great Vessel V53Gy (cc)	≤0.035	0.1875 ± 0.36	0.1305 ± 0.25	0.057	NA
Esophagus V35Gy (cc)	≤0.035	0	0.003 ± 0.006	−0.003	NA

* Statistical significance. NA: no meaningful test could be performed due to insufficient number of non-zero difference pairs.

## Data Availability

All data are available upon reasonable request and subject to institutional approval.

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
