# Peer review of "MR-Guided Radiation Therapy for Prostate and Pancreas Cancer Treatment: A Dosimetric Study Across Two Major MR-Linac Platforms"

_cancers, 2025, doi:10.3390/cancers17162708_

Round 1

Reviewer 1 Report

Comments and Suggestions for Authors

With the growing clinical implementation of MR-Linacs, this article comes as a suitable addition to the existing literature. The paper examines dosimetric aspects of treatment planning / delivery for prostate and pancreatic cancers, respectively, also evaluating delivery times and overall plan quality via gamma analysis. The work compares two MR-Linac platforms (Elekta Unity vs ViewRay MRIdian) regarding the above parameters.

This is a simple but useful dosimetric study, showing no statistically significant differences in terms of OAR doses between the two platforms. Overall, the study concludes that the two systems deliver dosimetrically comparable treatment plans over similar time periods for prostate and pancreatic cancer patients.

The paper is generally well written and the scientific component / statistical analysis are sound. I have a few relatively minor comments to improve the paper:

  1. Did the authors use any specific selection criteria for the two trials (prostate and pancreas) discussed in the paper? Please add a couple of sentences justifying the choice of the anatomical location. You mention that the MRIdian system has been used in two major clinical trials – yet the literature indicated towards several trials employing this MR-Linac, including lung, breast, brain.
  2. Please use the same number of decimals for the p value (tables).
  3. The limitations of this work (Discussion) should also include the retrospective nature of the study, the very small patient cohort (20 patients) and the limited anatomical locations (prostate and pancreas).
  4. Line 111 – replace ‘does’ with ‘dose’.

Author Response

With the growing clinical implementation of MR-Linacs, this article comes as a suitable addition to the existing literature. The paper examines dosimetric aspects of treatment planning / delivery for prostate and pancreatic cancers, respectively, also evaluating delivery times and overall plan quality via gamma analysis. The work compares two MR-Linac platforms (Elekta Unity vs ViewRay MRIdian) regarding the above parameters.
This is a simple but useful dosimetric study, showing no statistically significant differences in terms of OAR doses between the two platforms. Overall, the study concludes that the two systems deliver dosimetrically comparable treatment plans over similar time periods for prostate and pancreatic cancer patients.

The paper is generally well written and the scientific component / statistical analysis are sound. I have a few relatively minor comments to improve the paper:

Comment 1: Did the authors use any specific selection criteria for the two trials (prostate and pancreas) discussed in the paper? Please add a couple of sentences justifying the choice of the anatomical location. You mention that the MRIdian system has been used in two major clinical trials – yet the literature indicated towards several trials employing this MR-Linac, including lung, breast, brain.

Response: 

Both the prostate and pancreas are among the most commonly treated anatomical sites on MR-Linac platforms, largely due to the superior soft-tissue contrast provided by MRI, as well as the integration of real-time gating and online adaptive capabilities. These features are particularly advantageous for managing organ motion and minimizing dose to adjacent critical structures during prostate and pancreas treatment. We selected these two sites based on their clinical relevance and the robust body of evidence supporting their treatment on MR-Linacs. Specifically, the MIRAGE and SMART trials are two landmark studies representing prostate and pancreas SBRT, respectively, and exemplify the potential of MR-guided radiotherapy. Additionally, our institutions actively participated in both trials using the ViewRay MRIdian system. This provided a unique opportunity to benchmark clinical plan quality from MRIdian against that of Elekta Unity, facilitating a meaningful evaluation of Unity’s performance for these anatomic sites.

We acknowledge the reviewer’s suggestion and have added the following to the Introduction Section:
“Both the prostate and pancreas are among the most commonly treated anatomical sites on MR-Linac platforms, largely due to the superior soft-tissue contrast provided by MRI, as well as the integration of real-time gating and online adaptive capabilities. These features are particularly advantageous for managing organ motion and minimizing dose to adjacent critical structures during prostate and pancreas treatment, allowing for toxicity reduction and dose escalation.”

Comment 2: Please use the same number of decimals for the p value (tables).

Response:

We appreciate the reviewer’s comment, and have revised the table accordingly.

Comment 3: The limitations of this work (Discussion) should also include the retrospective nature of the study, the very small patient cohort (20 patients) and the limited anatomical locations (prostate and pancreas).

Response:

In response to the reviewer’s suggestion, we have expanded the Discussion Section to include these potential limitations:
“Finally, this is a retrospective study with limited number of patients, potentially limiting the statistical power. Moreover, only prostate and pancreas treatments have been investigated in this work. Evaluation across additional anatomical sites with daily anatomical variation using larger sample size and clinical endpoints could provide deeper insights into the unique advantages and challenges of each system, and should be studied in the future work.”

Comment 4: Line 111 – replace ‘does’ with ‘dose’.

Response: 

We apologize for the typo, and have revised manuscript accordingly.

Reviewer 2 Report

Comments and Suggestions for Authors

This study compares dosimetric performance between two major MR-guided radiotherapy platforms—ViewRay MRIdian and Elekta Unity—in treating prostate and pancreatic cancers. By re-planning 20 patient cases from MRIdian onto the Unity system using matched imaging and parameters, the authors found that both systems delivered comparable treatment quality. Statistically significant reductions in rectum and femur doses were observed in Unity for prostate cases, and Unity also showed a lower mean liver dose for pancreas cases, although most differences were not clinically significant. Unity plans were confirmed to be clinically deliverable with high gamma pass rates, suggesting both systems are effective for high-quality MRgRT. However, several sections of the manuscript require revision and clarification before it can be considered for publication.

Major:

  1. The Unity plans were created with access to the MRIdian plan quality reports. How might this non-blinded approach have biased the results in favor of the Unity platform? Please consider discussing this limitation in more detail.
  2. The MRIdian plans were created by board-certified dosimetrists while Unity plans were created by residents. Could this difference in experience have influenced plan quality? How do you account for this in your comparisons?
  3. While some dosimetric differences were statistically significant, their clinical significance appears limited. Could the authors discuss potential implications of these differences on toxicity outcomes?
  4. The Unity and MRIdian systems use different MLC designs. Can the authors elaborate on how these differences may affect dose conformity and modulation complexity in real-world clinical settings?

Minor:

  1. Please clarify whether IRB or ethics committee approval was obtained for the retrospective use of patient data. If the data were derived from previous clinical trials (MIRAGE and SMART), was a waiver for this specific analysis granted?
  2. Please ensure consistent use of the abbreviation for adaptive radiotherapy/radiation therapy throughout the manuscript. Specifically: In line 38, the term adaptive radiotherapy is introduced without abbreviation. It is recommended that you introduce and define the abbreviation here as adaptive radiotherapy (ART). In line 50, the term adaptive radiation therapy is used. To maintain clarity and consistency, please choose either adaptive radiotherapy or adaptive radiation therapy as the standard term and use it consistently throughout the manuscript. Once defined, ART should be used uniformly in all subsequent occurrences in the text.
  3. In Table 3, please consider replacing the written expressions ">=" and "<=" under the "Trial Goal" column with their corresponding mathematical symbols: ≥ and ≤.
  4. Currently, Table 3 includes two separate sets of dosimetric results—one for prostate SBRT (MIRAGE trial) and one for pancreas SBRT (SMART trial)—presented as two adjacent but structurally separate tables under the same heading. To improve clarity and organization, I recommend the authors either: Split them into two clearly labeled tables or Merge both datasets into a single comprehensive table.

Author Response

This study compares dosimetric performance between two major MR-guided radiotherapy platforms—ViewRay MRIdian and Elekta Unity—in treating prostate and pancreatic cancers. By re-planning 20 patient cases from MRIdian onto the Unity system using matched imaging and parameters, the authors found that both systems delivered comparable treatment quality. Statistically significant reductions in rectum and femur doses were observed in Unity for prostate cases, and Unity also showed a lower mean liver dose for pancreas cases, although most differences were not clinically significant. Unity plans were confirmed to be clinically deliverable with high gamma pass rates, suggesting both systems are effective for high-quality MRgRT. However, several sections of the manuscript require revision and clarification before it can be considered for publication.

Comment 1: The Unity plans were created with access to the MRIdian plan quality reports. How might this non-blinded approach have biased the results in favor of the Unity platform? Please consider discussing this limitation in more detail.

Response: 

The authors’ appreciate the reviewer’s suggestion. One of the key advantages of having access to the MRIdian plan quality reports is that the planners will have the knowledge about what each plan could potentially achieve dosimetrically during the process of creating Unity plans. Thus, those MRIdian plans could serve as a benchmark during the planning process, partially contributing to the similar dosimetric performance observed in the Unity plans.

Nevertheless, the key question investigated in this work was whether the Unity system would yield comparable plans to that of the MRIdian system in the two aforementioned trials given the differences in Linac hardware and TPS design. Our study successfully demonstrated similar dosimetric performance between the two systems despite the different system designs. 

In response to the reviewer’s comment, we have added additional discussion in the Limitations Section:
“Additionally, the plan quality reports of the MRIdian plans were not blinded to the Unity planners, and thus what each plan could potentially achieve dosimetrically was known to the Unity planners a priori. This knowledge may have guided the approach to generating Unity plans by setting a benchmark, partially contributing to the similar dosimetric performance observed between the two platforms.”

Comment 2: The MRIdian plans were created by board-certified dosimetrists while Unity plans were created by residents. Could this difference in experience have influenced plan quality? How do you account for this in your comparisons?

Response:

Difference both in planning and TPS experience may potentially affect plan quality especially given that the Unity Monaco TPS is new to the residents. In our study, this was accounted for via the supervision of three board-certified medical physicists with expertise in MRgRT planning on MRIdian (JL and XQ) and Unity (DH) platforms.

We have expanded the Limitations Section to include this information:
“First, the MRIdian and the Unity plans of the same patient were created by different planners. The clinical MRIdian treatments were generated by board-certified dosimetrists and/or physicists, whereas the Unity counterparts were planned by resident physicists after a short course of Monaco TPS training. To account for the difference in planning experience, Unity plans were supervised by three board-certified medical physicists with expertise in MRgRT planning on MRIdian (J.L. and X.Q.) and Unity (D.H.) platforms.”

Comment 3: While some dosimetric differences were statistically significant, their clinical significance appears limited. Could the authors discuss potential implications of these differences on toxicity outcomes?

Response: 

We acknowledge the reviewer’s suggestion. The minor dosimetric differences observed in this study will have minimal impact on clinical outcomes and toxicity rates.

Comment 4: The Unity and MRIdian systems use different MLC designs. Can the authors elaborate on how these differences may affect dose conformity and modulation complexity in real-world clinical settings?

Response: 

The authors acknowledge the reviewer’s comment. We have revised Section 4.3 Beam Shaping and Plan Quality in the revised manuscript:

“The Unity system elected a single stacked and single focused MLC design, whereas MRIdian utilized a double-stack, double-focused MLC with a smaller effective leaf width and shorter SAD. These differences in beam shaping between the two systems could potentially impact treatment plan quality in terms of dose conformity, reducing beam penumbra, and allowing steeper dose falloff. Interestingly, comparable plans were observed in our study between the two systems. This may be partially attributed to the dynamic jaw tracking of the Unity system along with the Monaco TPS which provides relatively more intuitive planning experience and advanced planning capabilities such as multi-criteria and biology-based optimization, compensating for the differences in MLC and machine SAD.”

Comment 5: Please clarify whether IRB or ethics committee approval was obtained for the retrospective use of patient data. If the data were derived from previous clinical trials (MIRAGE and SMART), was a waiver for this specific analysis granted?

Response: 

We have added IRB/ethics committee approval statement in the revised manuscript.

Comment 6: Please ensure consistent use of the abbreviation for adaptive radiotherapy/radiation therapy throughout the manuscript. Specifically: In line 38, the term adaptive radiotherapy is introduced without abbreviation. It is recommended that you introduce and define the abbreviation here as adaptive radiotherapy (ART). In line 50, the term adaptive radiation therapy is used. To maintain clarity and consistency, please choose either adaptive radiotherapy or adaptive radiation therapy as the standard term and use it consistently throughout the manuscript. Once defined, ART should be used uniformly in all subsequent occurrences in the text.

Response: 

We appreciate the reviewer’s feedback and have revised the manuscript accordingly.

Comment 7: In Table 3, please consider replacing the written expressions ">=" and "<=" under the "Trial Goal" column with their corresponding mathematical symbols: ≥ and ≤.

Response:

We appreciate the reviewer’s feedback and have revised the manuscript accordingly.

Comment 8: Currently, Table 3 includes two separate sets of dosimetric results—one for prostate SBRT (MIRAGE trial) and one for pancreas SBRT (SMART trial)—presented as two adjacent but structurally separate tables under the same heading. To improve clarity and organization, I recommend the authors either: Split them into two clearly labeled tables or Merge both datasets into a single comprehensive table.

Response: 

We appreciate the reviewer’s suggestion. We have divided table into two different tables in the revised manuscript.

Reviewer 3 Report

Comments and Suggestions for Authors

This paper compares dosimetric characteristics and delivery metrics of prostate and pancreas treatments on two major MR-Linac systems (ViewRay MRIdian vs. Elekta Unity), by retrospectively re-planning 20 clinical cases. The topic is interesting and clinically relevant given the growing adoption of MR-guided radiotherapy.

 I have some questions and suggestions below:

SPECIFIC OBSERVATIONS:  The manuscript does not include a “Simple Summary”, which is required by Cancers for clear communication of key findings to a broader clinical audience. There is also no explicit mention of ethical approval or IRB number, nor any statement on informed consent for the use of patient data. all of the standard sections that should appear at the end of the manuscript are missing. Specifically, there is no Author Contributions, Funding, Institutional Review Board Statement, Informed Consent Statement, Data Availability Statement, or Conflicts of Interest section.

Materials and methods: 

How exactly were these 20 patients selected? Was this truly random, or based on case availability? What were the inclusion and exclusion criteria

Was any power analysis performed to determine if 10 cases per site was sufficient to detect meaningful dosimetric differences?

Could the authors clarify how planning consistency was maintained, given that the Unity plans were created by different planners (resident physicists) and not blinded to the MRIdian plan reports? Could this have biased the results?

Given the known impact of motion management differences, why were no dynamic adaptation or gated delivery comparisons included, especially since pancreas cases are highly sensitive to intra-fraction motion?

Were the dose distributions and constraints independently reviewed by a radiation oncologist, or was this purely a physics planning study?

Discussion: It would be helpful for the authors to discuss how the observed minor dosimetric differences might or might not impact clinical outcomes or toxicity rates, given the MIRAGE and SMART trial benchmarks.

Author Response

This paper compares dosimetric characteristics and delivery metrics of prostate and pancreas treatments on two major MR-Linac systems (ViewRay MRIdian vs. Elekta Unity), by retrospectively re-planning 20 clinical cases. The topic is interesting and clinically relevant given the growing adoption of MR-guided radiotherapy.

 I have some questions and suggestions below:

Comment 1: SPECIFIC OBSERVATIONS:  The manuscript does not include a “Simple Summary”, which is required by Cancers for clear communication of key findings to a broader clinical audience. There is also no explicit mention of ethical approval or IRB number, nor any statement on informed consent for the use of patient data. all of the standard sections that should appear at the end of the manuscript are missing. Specifically, there is no Author Contributions, Funding, Institutional Review Board Statement, Informed Consent Statement, Data Availability Statement, or Conflicts of Interest section.

Response:

The authors acknowledge the reviewer’s comments, and have revised the manuscript to include all the missing sections.

Comment 2: How exactly were these 20 patients selected? Was this truly random, or based on case availability? What were the inclusion and exclusion criteria.

Response:

In this retrospective study, the 20 patients were selected from all the patients treated on the aforementioned trials at our institution from 2021 to 2023. The exclusion criterion was different prescription doses than the ones recommended by the respective trials. In response to the reviewer’s comment, we have added this information in the Materials and Methods Section.

Comment 3: Was any power analysis performed to determine if 10 cases per site was sufficient to detect meaningful dosimetric differences?

Response:

Power analysis was not performed for the current study. We acknowledge the potentially limited power, and have discussed this as a limitation of the study.

Comment 4: Could the authors clarify how planning consistency was maintained, given that the Unity plans were created by different planners (resident physicists) and not blinded to the MRIdian plan reports? Could this have biased the results?

Response:

We appreciate the reviewer’s question/comment. Indeed, difference both in planning and TPS experience may potentially affect plan quality especially given that the Unity Monaco TPS is new to the resident physicists. In our study, the planning consistency was maintained via the supervision of three board-certified medical physicists with expertise in MRgRT planning on MRIdian (JL and XQ) and Unity (DH) platforms.

We have expanded the Limitations Section to include this information:
“First, the MRIdian and the Unity plans of the same patient were created by different planners. The clinical MRIdian treatments were generated by board-certified dosimetrists and/or physicists, whereas the Unity counterparts were planned by resident physicists after a short course of Monaco TPS training. To account for the difference in planning experience, Unity plans were supervised by three board-certified medical physicists with expertise in MRgRT planning on MRIdian (J.L. and X.Q.) and Unity (D.H.) platforms.”

Comment 5: Given the known impact of motion management differences, why were no dynamic adaptation or gated delivery comparisons included, especially since pancreas cases are highly sensitive to intra-fraction motion?

Response:

The authors appreciate the reviewer’s suggestion. Investigations on plan adaptations and motion management between MRIdian and Unity are indeed intriguing, and could potentially benefit the audience of the MRgRT community. This preliminary study aims to address the main concern regarding how dosimetrically different system design (e.g., the MLC, SAD, etc.) could affect the characteristics of a treatment plan, and thus advanced topics such as motion management were not investigated.

Comment 6: Were the dose distributions and constraints independently reviewed by a radiation oncologist, or was this purely a physics planning study? 

Response:

All MRIdian clinical plans were reviewed by radiation oncologists, whereas all Unity plans were reviewed by the supervising medical physicists who are active on MRgRT service.

Comment 7: Discussion: It would be helpful for the authors to discuss how the observed minor dosimetric differences might or might not impact clinical outcomes or toxicity rates, given the MIRAGE and SMART trial benchmarks.

Response:

We acknowledge the reviewer’s suggestion. The minor dosimetric differences observed in this study will have minimal impact on clinical outcomes and toxicity rates.

Reviewer 4 Report

Comments and Suggestions for Authors

In this very interesting investigation entitled "MR-Guided Radiotherapy for Prostate and Pancreas Cancer Treatment: A Dosimetric Study Across Two Major MR-Linac Platforms", the authors randomly selected 20 MRIdian patients with prostate and pancreas cancers from the MIRAGE and SMART trials, and created a retrospective treatment plan on the Unity system, therefore performing a comparative dosimetric analysis between the two types of MR-linac systems.

In summary, In this investigation most organ-at-risk dose-volume metrics showed no statistically significant differences, and for prostate patients, Unity demonstrated lower rectum V36Gy, V38Gy, V40Gy, and lower left and right femur V20Gy ; whereas it showed lower mean liver dose for pancreas patients. Mean delivery times for prostate treatments were 12.78±1.68 min (MRIdian), and 13.53±1.88 min (Unity), and 14.58±2.78 min (MRIdian) and 17.40±3.77 min (Unity) for pancreas treatments. Overall, comparable treatment quality and delivery times were observed between the two platforms thus highlighting that both MR-Linac systems offer comparable capabilities for high-quality radiation therapy planning.

Although I am not an expert in this field, I did not identify any inaccurate or misleading information in this work. To my point of view, this work is significant to the field, the introduction provides sufficient background, and the research, and methods are adequately described. The results are clearly presented, and the conclusion supported by the results. I thus recommend publication in cancers, at MDPI, after the following minor revisions have been performed. These modifications are more intended to assist the authors by enhancing the overall quality, and readership of their manuscript.

‒ The abstract section does not fully align with an abstract paragraph. Certain sentences could benefit from being summarized, and/or removed for conciseness, and clarity.

‒ L111: doses

‒ Put a space after 2 and 3 of 2mm, and 3mm

‒ Put a space before and after = of difference=3%, (DTA)=2mm, etc.

‒ Every minus sign must be an en dash (insert symbol).

‒ dashes such as for instance L138: 1.26-1.87

‒ >= or <= must be replaced by ≥ or ≤

Author Response

In this very interesting investigation entitled "MR-Guided Radiotherapy for Prostate and Pancreas Cancer Treatment: A Dosimetric Study Across Two Major MR-Linac Platforms", the authors randomly selected 20 MRIdian patients with prostate and pancreas cancers from the MIRAGE and SMART trials, and created a retrospective treatment plan on the Unity system, therefore performing a comparative dosimetric analysis between the two types of MR-linac systems.
In summary, In this investigation most organ-at-risk dose-volume metrics showed no statistically significant differences, and for prostate patients, Unity demonstrated lower rectum V36Gy, V38Gy, V40Gy, and lower left and right femur V20Gy ; whereas it showed lower mean liver dose for pancreas patients. Mean delivery times for prostate treatments were 12.78±1.68 min (MRIdian), and 13.53±1.88 min (Unity), and 14.58±2.78 min (MRIdian) and 17.40±3.77 min (Unity) for pancreas treatments. Overall, comparable treatment quality and delivery times were observed between the two platforms thus highlighting that both MR-Linac systems offer comparable capabilities for high-quality radiation therapy planning.

Although I am not an expert in this field, I did not identify any inaccurate or misleading information in this work. To my point of view, this work is significant to the field, the introduction provides sufficient background, and the research, and methods are adequately described. The results are clearly presented, and the conclusion supported by the results. I thus recommend publication in cancers, at MDPI, after the following minor revisions have been performed. These modifications are more intended to assist the authors by enhancing the overall quality, and readership of their manuscript.

Comment1: The abstract section does not fully align with an abstract paragraph. Certain sentences could benefit from being summarized, and/or removed for conciseness, and clarity.

Response: 

The authors acknowledge the reviewer’s comments, and have revised the manuscript.

Comment 2: L111: doses.
Comment 3: Put a space after 2 and 3 of 2mm, and 3mm.
Comment 4: Put a space before and after = of difference=3%, (DTA)=2mm, etc.
Comment 5: Every minus sign must be an en dash (insert symbol).
Comment 6: dashes such as for instance L138: 1.26-1.87.
Comment 7: >= or <= must be replaced by ≥ or ≤.

Response to Comment 2-7:

We appreciate the reviewer’s suggestion, and have revised the manuscript accordingly.

Round 2

Reviewer 2 Report

Comments and Suggestions for Authors

The author has addressed the reviewers’ comments effectively, and the revised manuscript meets the publication standards. Therefore, I recommend it for publication.

Reviewer 3 Report

Comments and Suggestions for Authors

Dear Authors,
I appreciate your efforts in addressing the comments and improving the manuscript accordingly. Your careful attention to the feedback and the quality of your work are commendable. No more comments.